# Fluorogenic Detection of Human Serum Albumin Using Curcumin-Capped Mesoporous Silica Nanoparticles

**DOI:** 10.3390/molecules27031133

**Published:** 2022-02-08

**Authors:** Ismael Otri, Serena Medaglia, Elena Aznar, Félix Sancenón, Ramón Martínez-Máñez

**Affiliations:** 1Instituto Interuniversitario de Investigación de Reconocimiento Molecular y Desarrollo Tecnológico (IDM), Universitat Politècnica de València, Universitat de València, 46022 Valencia, Spain; isot@doctor.upv.es (I.O.); sermed@idm.upv.es (S.M.); 2Departamento de Química, Universidad Politécnica de Valencia, Camino de Vera s/n, 46022 Valencia, Spain; 3CIBER de Bioingeniería, Biomateriales y Nanomedicina (CIBER-BBN), 46022 Valencia, Spain; 4Unidad Mixta de Investigación en Nanomedicina y Sensores, Instituto de Investigación Sanitaria La Fe (IISLAFE), Universitat Politècnica de València, 46026 Valencia, Spain; 5Unidad Mixta UPV-CIPF de Investigación en Mecanismos de Enfermedades y Nanomedicina, Centro de Investigación Príncipe Felipe, Universitat Politècnica de València, 46100 Valencia, Spain

**Keywords:** mesoporous silica, gated hybrid materials, sensors, HSA detection

## Abstract

Mesoporous silica nanoparticles loaded with rhodamine B and capped with curcumin are used for the selective and sensitive fluorogenic detection of human serum albumin (HSA). The sensing mesoporous silica nanoparticles are loaded with rhodamine B, decorated with aminopropyl moieties and capped with curcumin. The nanoparticles selectively release the rhodamine B cargo in the presence of HSA. A limit of detection for HSA of 0.1 mg/mL in PBS (pH 7.4)-acetonitrile 95:5 *v*/*v* was found, and the sensing nanoparticles were used to detect HSA in spiked synthetic urine samples.

## 1. Introduction

Human serum albumin (HSA) is a vital protein that constitutes around 50% of the total proteins in human plasma [1]. HSA is synthesized in the liver, and it is related with the transport of several endogenous and exogenous biomolecules such as fatty acids, thyroxine, hormones and drugs [2]. Moreover, HSA also plays a vital role in the regulation of plasma osmotic pressure, maintain water equilibrium between tissues and preserve blood pH [3]. Normal levels of HSA in serum are in the 35–55 g/L range, whereas in urine its concentration is ca. 30 mg/L. However, high levels of HSA in body fluids are related with several human diseases such as kidney failure, diabetes mellitus, cardiovascular disorders, obesity and liver injury [4]. Moreover, low HSA levels are found to cause chronic hepatitis, liver failure and cirrhosis [5].

Currently, HSA is detected and quantified using immune-electrophoresis, enzyme-linked immunosorbent assays (ELISA), radio-immunoassays and liquid chromatography–mass spectroscopy [6,7]. These methods presented several drawbacks, such as the need of complicated instrumentation and the assistance of trained personnel, and are long-time procedures [8]. Recently, as an alternative to these classical methods, fluorescent molecular-based probes, polymers and gold nanoparticles have been described for HSA detection [9,10]. However, some of those methods suffer interferences from molecules in the human serum samples, such as hormones, growth factors, fats, carbohydrates and inorganic substances [11]. Taking into account the above-mentioned facts, the development of fast, easy-to-use, selective and reliable methods to detect HSA in biological matrices for clinical diagnosis and biomedical research is of relevance.

From another point of view, mesoporous silica nanoparticles (MSNs) functionalized with molecular gates able to provide on-command cargo release have been used to design smart nanodevices with application in the biomedical field [12,13,14], for example as drug-controlled release systems [15,16], for genetic material delivery [17,18], biosensing [19,20], bioimaging [21,22], tissue engineering [23,24], theragnostic applications [25], immunotherapy [26,27] or communication protocols [28,29,30,31]. Specifically, the development of new sensing/recognition protocols using mesoporous silica nanoparticles (MSNs) equipped with molecular gates has boosted and several interesting examples has been described in the literature [32]. In these sensing materials, the pores of the nanoparticles are loaded with a dye/fluorophore (acting as reporter) and the external surface is functionalized with bulky (supra)molecular ensembles, which disable the release of the entrapped reporter. In the presence of a target analyte, which selectively interacted with the gating (supra)molecular ensemble, pores opened, the dye/fluorophore is released and a macroscopic signal (changes in color or in the fluorescence) is finally generated [33,34]. Considering that most of the sensing protocols are designed to detect target analytes in water, the best shape of MCM-41 type mesoporous silica for its use in the preparation of gated sensory materials is nanoparticles of about ca. 100 nm diameter. This allowed the proper suspension of the gated material in aqueous environments and the effective recognition of the target analyte.

Taking into account our interest in the development of new sensing nanodevices to detect biomolecules [35,36], herein we present the synthesis and characterization of curcumin-capped mesoporous silica nanoparticles for the sensitive and selective detection and quantification of HSA. Mesoporous silica nanoparticles (ca. 100 nm diameter) were selected, as inorganic scaffold, and the pores loaded with rhodamine B as reporter. The outer surface of the loaded nanoparticles was decorated with aminopropyl moieties (which are positively charged at neutral pH). Finally, the pores were capped upon addition of curcumin. Nearly half of the curcumin molecules (which have three ionizable hydroxyl groups with p*K*_a_ values of 7.8, 8.5 and 9.0) [37] had a negative charge at physiological pH and, for this reason, yielded strong electrostatic interactions with the positively charged protonated amino moieties. The signaling paradigm relies on the fact that HSA shows a marked affinity for complexation with curcumin [38]. In the presence of HSA curcumin molecules are expected to be displaced from the surface of the nanoparticles, due to its preferential coordination with the protein, with subsequent pore opening and rhodamine B release (Figure 1). The increase in the emission intensity of rhodamine B in then solution would be directly related to the amount of HSA present in the medium. As far as we know, this is the first curcumin-capped hybrid nanomaterial used for the fluorogenic detection of HSA.

## 2. Results and Discussion

MCM-41 type MSNs were prepared using *n*-cetyltrimethylammonium bromide (CTABr) as template and tetraethylorthosilicate (TEOS) as a silica source [18,19] and then calcinated to remove the surfactant. MSNs were then loaded with rhodamine B, the loaded solid was reacted with (3-aminopropyl) triethoxysilane and, finally, the pores were capped by addition of curcumin (solid **S1** in Figure 1).

As-synthesized, calcined and **S1** nanoparticles were characterized using standard techniques such as powder X-ray diffraction (PXRD), transmission electron microscopy (TEM), thermogravimetric analysis, dynamic light scattering (DLS) and elemental analysis. Figure 1A shows the PXRD patterns of the as-made (curve a), calcined (curve b) and **S1** nanoparticles (curve c) which confirmed the mesoporous structure of the materials and the preservation of its structural features after the loading and functionalization process (for **S1**). Moreover, TEM images of calcined MSNs (Figure 1B, image d) and **S1** (Figure 1B, image e) show that nanoparticles were spherical with an average diameter of ca. 110 nm.

N_2_ adsorption–desorption isotherms of calcined MSNs showed a type IV curve. From BET and BJH models, a specific surface area of 1207.3 m^2^ g^−1^, a narrow pore size distribution and an average pore diameter of 2.56 nm for the starting calcined MCM-41 was obtained. After the loading and functionalization processes the surface area of **S1** markedly decreased to 17.6 m^2^ g^−1^. Structural parameters for calcined MCM-41 and **S1** are listed in Table 1. In addition, Z-potential of the starting calcined MSNs was negative (−26.1 mV) due to the presence of silanolate moieties on the surface of the nanoparticles (see Figure 1C). After loading of the pores with rhodamine B and functionalization of the external surface with aminopropyl moieties, the Z-potential became positive (36.4 mV) due to the presence of ionisable amino groups. The Z-potential of the final **S1** nanoparticles was reduced to (12.1 mV), ascribed to the capping of the pores with the negatively charged curcumin. Moreover, DLS measurements also showed an enhancement of the hydrodynamic diameter of the nanoparticles from 135 nm for calcined MSNs to 955 nm for **S1** (Figure 1D). The organic matter content in **S1** was 1.56 g/g SiO_2_ calculated by thermogravimetric analysis. Moreover, the amount of rhodamine B loaded in **S1** was estimated to be 0.18 g/g SiO_2_ using a calibration curve. To infer so, the mesoporous scaffold was hydrolyzed by incubating **S1** with NaOH 20% at 40 °C for 1 h and then the supernatant absorbance was measured.

After characterization, the cargo-controlled release studies in the absence and in the presence of HSA were carried out. In a typical experiment, solid **S1** (0.5 mg) was suspended in PBS (pH 7.4)-acetonitrile 95:5 *v*/*v* (1 mL) in the absence and in the presence of HSA (1000 μg mL^−1^). At selected times (0, 5, 15, 30 min) aliquots (120 μL) were collected and the supernatant was separated using a 0.22 μm filter. Then, rhodamine B emission at 571 nm (λ_ex_ = 555 nm) in the supernatant was measured. The obtained kinetic release profiles of rhodamine B from **S1** in the absence and in the presence of HSA are shown in Figure 2. As could be seen, in the absence of HSA a nearly zero release of rhodamine B was observed due to an efficient pore closure as a result of strong electrostatic interactions between the grafted positively charged aminopropyl moieties and the anionic curcumin. In a clear contrast, a marked rhodamine B release was observed (ca. 75% of the maximum amount of delivered fluorophore after 5 min) in the presence of HSA, which was ascribed to pore opening due to the formation of a complex between the HSA and curcumin (the logarithm of the stability constant for the HSA-curcumin complex is 4.74) [39].

Once the proper working of the gating mechanism in **S1** was assessed, rhodamine B release in the presence of increasing amounts of HSA was evaluated after 5 min upon addition (at this time 75% of the maximum amount of rhodamine B delivered was released from **S1** and this quantity is enough to produce a marked fluorescent signal for analytical purposes). The obtained results are shown in Figure 3. As could be seen, the addition of increasing amounts of HSA induced a clear emission enhancement at 571 nm ascribed to an enhanced rhodamine B released from **S1**. From the emission titration profile, a limit of detection as low as 0.1 mg/mL of HSA was determined (using 3SD/S, where SD is the standard deviation and S is the slope of the linear range). **S1** nanoparticles are stable at room temperature and did not require a special temperature for storage [40,41]. Another advantage of **S1** is the signal amplification observed [42]. In particular, it was confirmed that the presence of one HSA molecule (at a concentration of ca. 1.0 × 10^−5^ mol/L) results in the release of ca. 120 molecules of rhodamine B from **S1**.

Then, the selectivity of **S1** toward HSA was assessed by testing cargo release against common interfering molecules present in biological samples. Figure 4 shows the emission of rhodamine B (at 571 nm) released from **S1** nanoparticles suspended in PBS (pH 7.4)-acetonitrile 95:5 *v*/*v* in the presence of HSA and other selected interfering molecules such as anions, cations, amino acids, urea (at 10 μM) and synthetic urine in PBS [43]. As could be seen, only HSA was able to induce pore opening and rhodamine B release. These results indicated the high selectivity achieved with **S1** nanoparticles, as only the presence of HSA induced pore opening and rhodamine B release.

Finally, in order to test the applicability of **S1** to detect HSA in a complex biological environment, we evaluated the possible use of the sensing nanoparticles to determine HSA in synthetic urine. For this purpose, synthetic urine was spiked with known amounts of HSA and the concentration was determined using **S1** nanoparticles by means of a calibration curve in the same media. Results are shown in Table 2. As it can be seen, **S1** was satisfactorily applied to the detection of HSA with high recovery ratios in the 87–108% range. These results demonstrate the potential application of **S1** for the detection and quantification of HSA in realistic urine samples.

## 3. Materials and Methods

### 3.1. General Techniques

Powder X-ray diffraction (PXRD), transmission electron microscopy (TEM), N_2_ adsorption–desorption, thermogravimetric analysis (TGA) and fluorescence spectroscopy were employed to characterize the synthesized materials. PXRD measurements were performed on a D8 Advance diffractometer using CuKα radiation (Philips, Amsterdam, The Netherlands). Thermogravimetric analyses were carried out on a TGA/SDTA 851e balance (Mettler Toledo, Columbus, OH, USA), using an oxidizing atmosphere (air, 80 mL min^−1^) with a heating program: a gradient of 393–1273 K at 10 °C min^−1^, followed by an isothermal heating step at 1273 °C for 30 min. TEM images were obtained with a 100 kV CM10 microscope (Philips). N_2_ adsorption–desorption isotherms were recorded with an ASAP2010 automated adsorption analyser (Micromeritics, Norcross, GA, USA). The samples were degassed at 120 °C in vacuum overnight. The specific surface areas were calculated from the adsorption data in the low pressure range using the Brunauer, Emmett and Teller (BET) model. Pore size was determined following the Barret, Joyner and Halenda (BJH) method. Dynamic light scattering was used to obtain the particle size distribution of the different solids, using a ZetaSizer Nano ZS (Malvern Instruments, Malvern, UK) equipped with a laser of 633 nm and collecting the signal at 173°. For the measurements, samples were dispersed in distilled water. Data analysis was based on the Mie theory using refractive indices of 1.33 and 1.45 for the dispersant and mesoporous silica nanoparticles, respectively. An adsorption value of 0.001 was used for all samples. Variation of this adsorption value did not significantly alter the obtained distributions. Measurements were performed in triplicate.

### 3.2. Chemicals

Tetraethylorthosilicate (TEOS), *n*-cetyltrimethyl ammonium bromide (CTABr), curcumin, sodium hydroxide (NaOH), rhodamine B, tris(hydroxymethyl) aminomethane (TRIS), (3-aminopropyl) triethoxysilane, HSA, L-alanine, D-alanine, histidine, phenyl glycine, serine, valine, glutamine, and urea were purchased from Sigma Aldrich (Burlington, MA, USA). Curcumin was purchased from Tokyo Chemical Industry Co., Ltd. (TCI, Tokyo, Japan). Analytical-grade solvents were from Scharlab (Barcelona, Spain). All products were used as received.

### 3.3. Synthesis of Mesoporous Silica Nanoparticles (MSNs)

*n*-cetyltrimethylammonium bromide (CTABr, 1.00 g, 2.74 mmol) was first dissolved in 480 mL of deionized water. Then a 3.5 mL of NaOH 2 M in deionized water was added to the CTABr solution, followed by adjusting the solution temperature to 80 °C. TEOS (5 mL, 25.7 mmol) was then added dropwise to the surfactant solution. The mixture was allowed stirred for 2 h to give a white suspension. Finally, the solid was centrifuged, washed with deionized water and dried at 60 °C (MSNs as-synthesized). To prepare the final mesoporous material, the as-synthesized solid was calcined at 550 °C in oxygen atmosphere for 5 h to remove the template phase.

### 3.4. Synthesis of **S1**

In a typical synthesis, 750 mg of template-free MCM-41 were suspended in a solution of 340 mg of rhodamine B dye in 10 mL of miliQ water in a round-bottomed flask, (0.8 mmol of dye/g MCM-41). After 24 h stirring at room temperature, (3-aminopropyl) triethoxysilane (15 mmol/g) was added and the mixture was stirred for 6 h at room temperature. Then, curcumin (2.3 mmol/g) was added to the suspension. This suspension was stirred for 1 h at room temperature. Finally, this solid was filtered and washed with PBS to remove the unreacted alkoxysilane and the dye remaining outside the pores. The final solid **S1** was dried under vacuum at ambient temperature for 12 h.

### 3.5. Controlled Release Studies

Solid **S1** (0.5 mg) was suspended in PBS (pH 7.4)-acetonitrile 95:5 *v*/*v* (1 mL) in the absence and in the presence of HSA (1000 mg mL^−1^) and dye release studies versus time were performed at room temperature (25 °C) under stirring. At a certain time (0, 5, 15, 30 min), aliquots (120 µL) were collected, and the supernatant was separated using a 0.22 µm filter. The delivery of the rhodamine B dye was then monitored by its fluorescence emission band at 571 nm (λ_ex_ = 555 nm).

### 3.6. Synthetic Urine Preparation

Artificial urine was prepared by dissolving urea (18.2 g), potassium chloride (4.5 g), sodium chloride (7.5 g), sodium phosphate (4.8 g), and creatinine (2 g) in distilled water (750 mL). Then, the pH of the solution was adjusted to 6.

### 3.7. Calibration Curve with HSA

In a first step, **S1** (0.5 mg) was suspended in 1 mL of synthetic urine diluted with PBS (pH 7.4)-acetonitrile 95:5 *v*/*v* (10%). Then, increasing amounts (0.1 to 100 mg/mL) of HSA were solubilized in synthetic urine diluted with PBS (pH 7.4)-acetonitrile 95:5 *v*/*v* (10%). Then, the solutions of HSA were added to **S1** suspensions. After 5 min aliquots were taken and filtered using 0.2 µm filters. The delivery of rhodamine B dye was then monitored by the fluorescence emission band at 571 nm (λ_ex_ = 555 nm).

### 3.8. Selectivity Studies

**S1** (1.5 mg) was suspended in 3 mL PBS (pH 7.4)-acetonitrile 95:5 *v*/*v*. Then, the selected interfering molecules (anions, cations, amino acids, urea) were solubilized in artificial urine diluted with PBS (pH 7.4)-acetonitrile 95:5 *v*/*v* (10%) at a 10 mM concentration. Then, the selected interfering molecules (10 µM) and HSA (10 µM) were added to **S1** suspensions. After 5 min aliquots were taken and filtered using 0.2 µm filters. The delivery of rhodamine B dye was then monitored by the fluorescence emission band at 571 nm (λ_ex_ = 555 nm).

### 3.9. HSA Determination in Synthetic Urine

In a first step, **S1** (0.5 mg) was suspended in 1 mL of artificial urine diluted with PBS (pH 7.4)-acetonitrile 95:5 *v*/*v* (10%). Then, known amounts of HSA were solubilized in synthetic urine diluted with PBS (pH 7.4)-acetonitrile 95:5 *v*/*v* (10%). Then, solutions of HSA were added to **S1** suspensions. After 5 min aliquots were taken and filtered using 0.2 µm filters. The delivery of rhodamine B dye was then monitored by the fluorescence emission band at 571 nm (λ_ex_ = 555 nm).

## 4. Conclusions

In summary, we describe herein a sensing material based on gated mesoporous silica nanoparticles for the selective and sensitive HSA detection in buffered aqueous solution and in artificial urine. **S1** nanoparticles are composed of a mesoporous inorganic scaffold loaded with rhodamine B, with the external surface decorated with aminopropyl moieties and with the pores caped with curcumin. The sensing mechanism arises from a displacement reaction by the formation of a strong complex between HSA and curcumin that results in uncapping of the nanoparticles and rhodamine B release. A limit of detection for HSA of 0.1 mg/mL in PBS (pH 7.4)-acetonitrile 95:5 *v*/*v* was determined. It was also demonstrated that **S1** can be used to determine the concentration of HSA in spiked synthetic urine samples with recoveries in the 87–108% range.

## Data Availability

Not applicable.

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
