# Peer review of "Fluorogenic Detection of Human Serum Albumin Using Curcumin-Capped Mesoporous Silica Nanoparticles"

_molecules, 2022, doi:10.3390/molecules27031133_

Round 1

Reviewer 1 Report

The authors has described a novel and robust method to detect HSA using circumin-immobilised MSNs by trapping RhB for fluorometric detection. The concept and analytical performance and optimisation seems to be proven. However, some issues might need to be considered and amend as described below:

Please show the approximate amount of Rhodamine B that could be trapped into the MSNs (i.e., mol/particles)

From figure 2 and main text, it is confusing how the authors could calculate 100% of RhB concentration? As described that RhB can be simply trapped into the MSNs previously, thus regarding the nature of molecular dispersion and Fick’s law, it shouldn’t be likely to release at high amount up to or close to 100%

Though incubating is well investigated, it was not cleanse why 5 min was chosen to incubate the reaction. Explanation is needed and discussion.

Please show the plateau at the highest concentrations that this system can detect HSA in Figure 3, and show fitted equation. In addition, despite using HSA concentration at X axis, using log HSA concentration could display the linear range of the calibration curve and better to calculate LoD with 3SD/slope.

“S1 nanoparticles are stable at room temperature and did not require a special temperature for storage. Another advantage of S1 is the signal amplification observed. “ please add references to support these sentences as well as performing experiments to prove it.

Please explain why those ions and molecules in Fig4 were chosen. If they are related to human serum or fluid, please explain and also test the system in biological fluid.

It requires ethical permission to use urine as sample, please provide necessary information. Unless artificial urine was used, then provide where to purchase and resources.

Reviewer 2 Report

According to my opinion this paper is not presented well for this journal. Authors will clearly have to describe the methods and results. It will be better to add more Figs. or graphs to this paper. Overall this paper needs extensive revision. Highlight novelty of your paper in clear and concise manner. 

This paper should be rejected for the publication and authors should be given the opportunity to make extensive revision before resubmission.

Round 2

Reviewer 1 Report

The manuscript has been significantly improved, thus it is ready to be published.

Reviewer 2 Report

According to my opinion authors have not made significant changes in the manuscript. However, they must be given another opportunity to improve the results and the quality of the presentation. Specially the Figs. quality is not good and it seems that they just copied and pasted from somewhere. 

Please focus on the previous comments which are mentioned in my previous review. Quality of presentation is also needed to be improved.